

# PHYSICAL SOIL QUALITY INDICATORS FOR MONITORING BRITISH SOILS

Corstanje, Ron.[a,*], Mercer, Theresa G.[b], Rickson, Jane R.[a], Deeks, Lynda K.[a], Newell-Price, Paul[c], Holman, Ian[d], Kechavarsi, Cedric[e] and Waine, Toby W.[a]

[a] Cranfield Soil and Agrifood Institute, School of Water, Energy and Environment (SWEE), Cranfield University, Bedfordshire, MK43 0AL, UK

[b] Cranfield Institute for Resilient Futures, School of Water, Energy and Environment (SWEE), Cranfield University, Bedfordshire, MK43 0AL, UK

[c] ADAS Gleadthorpe, Meden Vale, Mansfield, Notts., NG20 9PF, UK

[d] Cranfield Water Science Institute, School of Water, Energy and Environment (SWEE), Cranfield University, Bedfordshire, MK43 0AL, UK

[e] Department of Engineering, University of Cambridge, University of Cambridge, Trumpington Street, Cambridge CB2 1PZ, UK

Corresponding author: t.mercer@cranfield.ac.uk

**Keywords:** Physical indicators, Soil quality, Monitoring, Proxies, Soil function

## Abstract

The condition or quality of soils determines its ability to deliver a range of functions that support ecosystem services, human health and wellbeing. The increasing policy imperative to implement successful soil monitoring programmes has resulted in the demand for reliable soil quality indicators (SQIs) for physical, biological and chemical



soil properties. The selection of these indicators needs to ensure that they are sensitive
and responsive to pressure and change e.g. they change across space and time in relation
to natural perturbations and land management practices. Using a logical sieve approach
based on key policy-related soil functions, this research assessed whether physical soil
properties can be used to indicate the quality of British soils in terms of its capacity to
deliver ecosystem goods and services. The resultant prioritised list of physical SQIs
were tested for robustness, spatial and temporal variability and expected rate of change
using statistical analysis and modelling. Six SQIs were prioritised; packing density, soil
water retention characteristics, aggregate stability, rate of erosion, depth of soil and soil
sealing. These all have direct relevance to current and likely future soil and
environmental policy and are appropriate for implementation in soil monitoring
programs.

## 33    1    Introduction

In recent years soil quality and its measurement have increasingly been based on soil
functions (Loveland & Thompson, 2002; Ritz *et al.*, 2009; Rosa, 2005). These functions
determine the ability of a soil to deliver and support ecosystem goods and services,
which have been linked to human health and wellbeing. Soils are typically recognised
for their role in provisioning goods such as building material, fresh water, fuel, fibre and
food (Robinson *et al.*, 2013). They also interact with other environmental components
(air and water), help preserve historic artefacts and burial grounds, and provide a
platform for infrastructure. The ecosystem services that rely on these functions include
regulation of climate and hydrology, contaminant transformation, biocontrol of plant
pathogens and parasites (Sylvain & Wall 2011) and water filtration/runoff
reduction/purification (Breure *et al.*, 2012). Supporting services provided by soils



include soil formation, soil fertility, biogeochemical cycling (C storage and nutrient
cycling), decomposition of organic materials and plant available water. A number of
cultural services are also supported such as recreational surfaces (Robinson *et al.*,
2013). In order to measure soil quality and function, soil quality indicators are
commonly used.
Indicators of soil quality are required for environmental monitoring/reporting and
provide the basis for many soil protection policies and monitoring programs (Pulleman
*et al.*, 2012). They help assess human and natural impacts on soils and to identify the
effectiveness (or otherwise) of sustainable land management practices (Doran & Parkin,
1994; Karlen & Stott, 1994; Schipper & Sparling, 2000). In order to assess soil quality,
a combined approach is required in which the biological, chemical and physical
attributes and their interactions are assessed (Bone *et al.*, 2010; Seybold *et al.*, 1998).
In this respect, monitoring is defined as a method to determine the quality and condition
of the soil environment over time. This is measured by determining actual values of the
attributes of interest.
There have been few studies that have discussed and attempted to prioritise the most
appropriate SQIs for biological (Masto *et al.*, 2015; Pulleman *et al.*, 2012; Ritz *et al.*,
2009) and physico-chemical indicators (Arshad & Coen, 1992; Asensio *et al.*, 2013;
Karlen & Stott, 1994; Masto *et al.*, 2015; Rickson *et al.*, 2012). This study focusses on a
systematic process of selection for physical SQI's and then explores their potential for
use in national monitoring schemes (e.g. England and Wales - (Loveland & Thompson,
2002; Merrington *et al.*, 2006), in particular exploring practical aspects such as





sampling design and size, the use of proxy's and pedotransfer functions and the
application of sensor technology.
The definition of what role or function a soil system should take can differ depending
on the stakeholder/user and their objectives (e.g. production, regulation or cultural)
(Rickson *et al.*, 2012). As such, indicators are usually selected on the basis of the
function(s) of interest, and observed and measured to infer the capability of a soil to
perform that function (Bone *et al.*, 2010; Ditzler & Tugel, 2002; Doran & Parkin, 1994).
Once selected, effective indicators need to meet the following criteria:
• Be meaningful, interpretable and sensitive (and measureable) to natural and
human induced pressures and change (Burger & Kelting, 1999; Loveland &
Thompson, 2002);

• Reflect the desired condition or end point for a particular soil and/or land use
and/or function (Loveland & Thompson, 2002);

• Be relatively cheap, practical and simple to monitor (Loveland & Thompson,
2002);

• Be responsive to corrective measures (Burger & Kelting, 1999);
• Be applicable over large areas and different soils/land use types (Burger &
Kelting, 1999);

• Be capable of providing continuous assessment over long time scales (Burger &
Kelting, 1999).

Selected physical SQIs need to be sensitive to pressure and reflect change in soil quality
status (the capacity of the soil to function) at any given location and time (Burger &
Kelting, 1999; Loveland & Thompson, 2002; Rickson *et al.*, 2012). As such, an
effective physical SQI would need to detect 'meaningful change' in a given soil





function and be responsive to this change in the light of expected changes in soil
quality. In other words, does the physical SQI change sufficiently that it can be
detected, and is this change indicative of a significant loss/gain in soil quality? In order
to evaluate the effectiveness of the indicator, the criteria for what constitutes a
'meaningful change' need to be set.
In some instances, where the indicator itself may drive the change (for example the
effect of bulk density on crop growth), 'meaningful change' may be as simple as
ascertaining the SQI value at a particular location and comparing this to a critical value
or target value or range. This approach is taken by Merrington et al. (2006) and whilst
simple in its approach, it does not capture the dynamic relationships between SQI and
soil functions. These relationships may differ between soil functions, land uses and soil
types (Jones, 1983). As such, there needs to be a focus on the dynamic relationships
between soil functions and SQIs: however, information in the literature is sparse.
Physical SQIs also need to be meaningful in terms of the soil processes that they
represent. A change in the SQI needs to relate to a change in the processes that are
taking place in the soil and therefore how the soil functions. For example, a change
(increase) in bulk density would result in a change in processes operating in the soil
(e.g. restriction to root elongation) and therefore a change in soil function (reduced crop
yield).
Soil properties are spatially and temporally variable as a result of land use and
management, parent material and climate. This variability introduces 'noise' into the
signal response (signal:noise i.e. meaningful change) in two ways. Firstly, there is the
consideration of the spatial unit over which the soil quality is assessed. The spatial



variability within this unit (e.g. plot, field, farm, catchment, national scale) will
introduce variability to the SQI, irrespective of whether there are any changes in soil
function(s). Secondly, there is the consideration of the impact of a particular land
management practice on the effectiveness of an SQI to indicate soil quality.
Based on the above criteria and considerations, it has been argued that it may not be
possible to achieve a single, affordable, workable soil quality index (Sojka & Upchurch,
1999) or a consensus on a standardised methodology which would be appropriate across
different soil and land use types (Karlen & Stott, 1994). Furthermore, soils can
frequently perform several functions simultaneously, although these can be diverse and
often conflicting, but must still be taken into account (Bone *et al.*, 2010; Schoenholtz *et*
*al.*, 2000).
This study uses a multi-stage approach in the selection and prioritisation of physical
SQIs that meet the required criteria and conditions outlined above. It consists of a
systematic review and selection procedure, followed by assessment of the selected
SQI's and how they could be best applied at a National scale monitoring programme.
The final priority list should indicate the soil's capacity to deliver ecosystem
goods/services and are therefore indicative of soil quality.

## 131    2    Methodology

The process of physical SQI selection takes a multi-stage approach as outlined in Figure
1. In the first stage, potential physical SQIs were identified from the available literature,
including those defined by Loveland and Thompson (2002) and Merrington et al.
(2006). Other physical SQIs (and the methods used to measure them) that had not been
considered previously were also included to produce an up-to-date list. In the second





stage, the candidate physical SQIs were prioritised using a logical sieve (Ritz *et al.*,
2009) and a scenario-based approach. In this approach, the logical sieve was
interrogated by running three scenarios based on typical priorities of different
stakeholders by applying weightings to the scores. As such, the approach was be used to
prioritise a specific soil function or degradation process of interest (Rickson *et al.*,

2012).

Finally, the priority physical SQIs were tested for robustness (statistical reliability and
accuracy as well as practicability), spatial and temporal variability and expected rate of
change using statistical analysis and modelling. This involves determining appropriate
sample numbers for defining meaningful change as well as proxy methods that can be
used to make the physical SQI measurements operational and feasible. For example,
where a standard measurement physical SQI measurement may be time or resource
intensive to measure and monitor in a large scale monitoring programme, an
easier/cheaper to measure proxy may exist that could make that physical SQI feasible
for inclusion into such a programme.
**2.1    Identification of potentially meaningful physical SQIs**
The identified physical SQIs were derived from the literature with consideration of
recent scientific advances and developments in soil policy. Loveland and Thompson
(2002) identified 22 direct and 4 indirect physical SQIs. Direct indicators refer to those
that are associated directly with a soil function, whereas indirect indicators refer to those
that are indirectly related to a soil function. Merrington et al. (2006) give the example of
soil water storage following rainfall as a direct indicator, whereas a catchment
hydrograph is an example of an indirect indicator of rainfall interception and storage by





soils. The initial physical SQIs were subsequently evaluated by Merrington et al.
(2006), resulting in 30 direct and 4 indirect physical indicators. A list of these is
provided in Table S1. Where an indicator could be measured using alternative
techniques/approaches, sub-categories reflecting this were created, ensuring that the
indicator and its different measurement methods were scrutinised by the logical sieve.
In this way, a total of 42 physical soil quality indicators were identified.
**2.2    Prioritisation of candidate physical SQIs**
The 42 physical SQIs were evaluated in terms of the following criteria:
•    **Criteria 1. Soil function**: does the candidate SQI reflect all soil function(s)? In

this case, the four main functions, as described in the Millennium Ecosystem

Assessment (Millennium Ecosystem Assessment, 2005), were used

(provisioning, regulation, cultural and supporting).

•    **Criteria 2. Land use**: does the candidate SQI apply to all land uses found

nationally? The range of land uses considered was based on the Centre for

Ecology and Hydrology's land cover map (CEH, 2007) that also reflected

differences in land use resulting from differences in land management practices

(e.g. cultivations on arable land as opposed to pasture).

•    **Criteria 3. Soil degradation process**: can the candidate SQI express soil

degradation processes? The range and representation that each physical SQI

gives to the main soil degradation processes as identified in the Thematic

Strategy for Soil Protection (European Commission, 2006, Table 4) was

considered. This approach captures whether the SQIs reflect the effect of

potential degradation threats on soil functions.

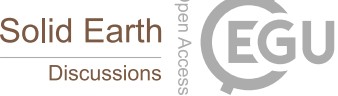

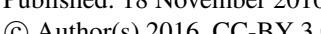

- **Criteria 4. Challenge criteria**: Does the candidate SQI meet the challenge criteria used by Merrington et al. (2006)? For example, is the indicator relevant to the function of the environmental interaction? Are the measurements of the indicator practicable? Can the indicator be measured cost effectively? Is the indicator policy relevant? These challenge criteria were developed for a national scale soil-monitoring scheme and were integrated with criteria used to identify the inverse of soil quality indicators from the ENVASSO project (Huber *et al.*, 2008).

Each of these criteria categories (and constituent factors) were considered separately and each of the physical SQI was scored numerically, with weighting factors using the approach outlined by Ritz et al (2009). The criteria are presented in Tables S2 – S5; the methodology for weighting, scoring and ranking in Methods S6; and an example of the logical sieve assessment in Table S7. Three scenarios were run to test the logical sieve.

Scenario 1 involved no weightings applied (all factors are equally important). For example, when considering Category 1 (soil functions category), all functions (factors) are equally important. In scenario 2 a higher priority was applied to the provisioning and regulation soil functions (factors). These two soil functions were selected as they are considered high priorities in current soil policy as highlighted in the Natural Environment White Paper, The Natural Choice (DEFRA, 2011a), the Soils Evidence Plan (DEFRA, 2011b) and the Welsh Soils Action Plan (Welsh Assembly Government, 2008). Scenario 3 used a weighting factor to normalise values across all categories. As such, differences in the number of factors in each category would not affect the outcome





(e.g. Category 1 (Soil Function) includes 4 factors to consider, whereas Category 2
(Land Use) has 7 factors to consider and so on).

These scenarios represent the types of questions that may be asked by different
stakeholder groups. The results from the three scenarios (top 25% cumulative scores, as
well as any of the physical SQIs that survived the sieving process by scoring > 0 in all
factors of all categories) resulted in 18 candidate physical SQIs. These were then further
filtered by the project team in the Rickson et al. (2012) study to ensure the results of the
logical sieve exercise were sensible and no indicators were disqualified unduly, data
were available to test the robustness of the selected SQIs, duplication/surrogacy and
scale issues (i.e. upscaling) were considered. The selected physical SQIs were reduced
to 7, based on whether there was scientific evidence regarding:

1.  What is the candidate SQI indicative of (i.e. what function is being degraded)?

2.  What is it responsive to? How responsive is it? (i.e. sensitivity, responsiveness)

3.  What factors may mitigate or accentuate the response (i.e. soil type, land use)?

4.  Is this indicator a first order indicator (i.e. a direct measure of the change in soil

quality) or a second, third, etc. order indicator (i.e. an indirect measure of the

change in the SQI, such as by remote sensing)

5.  Are there existing or suspected data-holdings for each indicator?

6.  How is it measured?

7.  What sampling support does it need?

8.  What is the sampling intensity required?



The final physical SQIs where such evidence was available that were further analysed

included:

- Packing density/bulk density

- Soil water retention characteristics

- Sealing

- Depth of soil

- Visual soil evaluation

- Rate of erosion

- Aggregate stability


## 2.3 Assessment of priority physical SQIs

These final physical SQIs were tested for uncertainty in their measurement, the spatial
and temporal variability in the indicator (as given by observed distributions) and the
expected rate of change (for a given soil function in light of expected changes in soil
quality) in the indicator. For each SQI, the following points were addressed:

- Whether the SQI could be directly related to to soil functions;

- What constitutes meaningful change in the SQI by determining the relationships
  between the SQI (and how it changes) and soil processes;

- The spatial variability of the SQI and the implications for sampling using spatial
  statistics and power analyses;

## 2.4 Statistical analyses and modelling approach

The type of analyses conducted on the SQIs depended on the type of soils data
available. Where full data were available, quantitative methods such as power analysis



or pedo-transfer functions were used. Otherwise analysis was carried out (semi)
qualitatively (e.g. using remote sensing) or qualitatively (where no data exists). Three of
the selected priority physical SQIs (Table 1) will be discussed. These represent
evaluations based on quantitative or semi-quantitative methods.
Where there was substantive quantity of data (e.g. packing density), we explored the
sampling intensity required to detect a change in the SQI. In other words, for the SQI to
be effective as an indicator, it needs to be sensitive to changes in soil quality, and
sufficiently responsive so to be detectable above the natural variability of the soil
(meaningful change) without requiring an impractical number of samples to determine
this change. We estimated the natural variability of a particular property in two ways; i)
through a natural stratification by land use and soil types and ii) through geostatistics
(see Methods S8), where we used block kriging to estimate the within block variance of
blocks sized 5, 10, 25 and 50 km$^2$, roughly approximating management unit of
increasing size (field, farm, landscape).

Where the particular property is obtained from complex analytical methods, such soil
water retention characteristics, we explored the use of pedotransfer functions, in
particular a multiple regression model and Multiple Additive Regression Splines, which
are described in Methods S9.

## 3    Results and Discussion
### 3.1    Packing Density
Packing density is a measure of soil porosity and an indirect measure of soil functions
such as water regulation, biomass production and habitat support. It also provides a



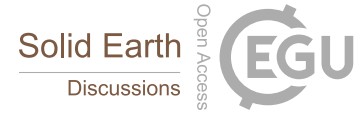

good estimate of soil compaction due to reduced total porosity. Compaction is generally
associated with land degradation (inverse of soil quality (Huber *et al.*, 2008)) and can
result in decreases in water holding capacity, water infiltration, microbial functions and
biogeochemical cycling (Edmondson *et al.*, 2011; Gregory *et al.*, 2015a). It is derived
by measuring dry bulk density (BD) modified by clay content (C) and is a very useful
parameter for spatial interpretations that require a measure of the compactive state of
soils (Jones *et al.*, 2003).
Bulk density (from which packing density is derived) is most commonly measured
using a Kopecky ring. This method is easy, convenient and cheap, but results can be
unrepresentative over large spatial areas due to the small diameter of the ring or
cylinder, and depth of measurement (usually 5cm). A number of proxies exist that
overcome some of the issues regarding sampling effort using the traditional Kopecky
ring method. These allow a higher resolution of measurements (1500-2500 samples) per
hectare over larger areas and include on-line (mobile) and non-mobile systems (Rickson
*et al.*, 2012). The methods used require multiple sensors and advanced techniques for
data analysis (Mouazen & Ramon, 2006) such as a combination of Visual and Near
Infrared (vis-NIR) measurements, combined with Theta probe determinations for soil
moisture or with soil resistance (penetrometer measurements) and vis-NIR
measurements to determine BD (and thus PD, when combined with clay content).
Measurements of packing density (PD) can detect relatively large changes in soil
physical properties. It has been used to detect differences in soil compaction between
different management practices, such as contrasting tillage systems (da Silva *et al.*,
2001; Dam *et al.*, 2005). For example, in no-till systems, BD can be 10% higher





compared to conventional tillage systems, particularly in the 0-10 cm layer (Dam *et al.*,

301    2005).

The power analysis based on land use by soil strata from national data (Figure 2) clearly
demonstrates the trade-off between the sample size required to detect change in packing
density (i.e. a change that impacts on soil functioning). Approximate sample sizes for a
national monitoring program can be determined based on expense and desired power. In
terms of sampling effort, it suggests that a different sampling regime would be required
for different geographical areas to ensure statistical robustness, taking into account the
different land use/soil and climate combinations.
The influence of spatial scales on sample size was calculated using a model-based
approach where the variation of different regions (size of spatial unit) was obtained
from a variogram in Methods S8. The sample size needed if a change is to be
determined over different spatial scales areas (i.e. field; 5 and 10 km$^2$; farm 25 km$^2$ and
landscape level 50 km$^2$) was determined (Figure 3). The graphs suggest that as spatial
area increases, the number of samples also needs to increase in order to determine
change within a given size of spatial area. If other factors that contribute to spatial
variability of PD (such as land use) are included, fewer samples are required.
**3.2   Soil Water Retention Characteristics**
Soil water retention characteristics (SWRC) encapsulate a number of important
capacity-based physical SQIs including plant available water capacity (PAWC), air
capacity (AC), relative field capacity (RFC), macroporosity (M), soil porosity
(Reynolds *et al.*, 2002, 2009) and the soil physical quality index Dexter S value (Dexter,
2004a, 2004b, 2004c). The Dexter S value is a measure of the micro-porosity of the soil



(Dexter, 2004c) and has been linked to a number of soil physical processes and soil
quality indicators, including bulk density. It is also related to root growth in soil
(Dexter, 2004a). Generally, the higher the value of S, the higher the soil physical
quality. It is recommended that the S value be used in combination with other capacity-
based indicators. This is because in some soils, values may be overestimated (e.g. sands
with unimodal and narrow pore size distributions) (Reynolds *et al.*, 2009).
Of these, PAWC, M and Dexter S value are related to root growth and therefore directly
to provisioning soil functions such as crop production. PAWC refers to the soil's
capacity to store and provide water that is available for uptake by plant roots. M
represents the volume of macropores with an equivalent pore diameter $\geq 300$ μm,
indicating the capacity of the soil to drain excess water quickly and facilitate root
growth (Reynolds *et al.*, 2009). RFC represents the proportion of pores filled with water
at field capacity and indicates the capacity of the soil to store water and air, relative to
the total pore volume.
The Dexter S value and other capacity-based physical SQIs are related to pore volume
and pore size distribution (Reynolds *et al.*, 2009). They are derived from soil hydraulic
behaviour and therefore are likely to be more sensitive to temporal and spatial changes
in soil condition and soil quality compared to other less dynamic indicators which look
solely at pore volume such as bulk density (Dexter, 2004a; Merrington *et al.*, 2006;
Naderi-Boldaji & Keller, 2016). The optimum values for each of the relevant physical
SQIs for the provisioning function are displayed in Table 2 and are assumed to
represent a meaningful change in the physical SQI as changes of this magnitude are
expected to affect root (and therefore crop) growth.



In order for the soil water retention characteristics SQI to be meaningful, it needs to be
indicative of soil functions that operate at different spatial scales (i.e. laboratory to field
to catchment). However, O'Connell et al. (2004, 2007) and Beven et al. (2008) discuss
uncertainties and inconsistencies in the measurement of rainfall and flow data between
years, which tend to dominate over the impacts of land use and management change on
flow characteristics at the catchment scale over time. These include uncertainties in
estimates of precipitation inputs to a catchment, uncertainty in measurements of stream
discharges (particularly during flooding events), and the uncertainty in characterising
land use / management patterns in space and time. Also, significant impacts at the small
scale may not have significant impact at catchment scales, due to landscape connectivity
(Rickson *et al.*, 2012). As such, there are gaps in connecting soil hydrological processes
and the physical properties that influence them at the larger scale, and this influences
any sampling efforts.
In terms of sampling effort, the standard Soil Survey of England and Wales method for
determining soil water retention characteristics is to collect three undisturbed soil
samples per horizon in winter or spring when the soil is near field capacity (Avery &
Bascomb, 1982). This involves using a coring device that reduces compaction during
sampling. The laboratory measurement of soil water retention characteristics can be
lengthy and requires considerable effort. The process involves saturation of the soil
samples, allowing soils to reach equilibration, determining bulk density and finally
calculating the volumetric water content at different soil water suctions. For the current
analysis, soil water retention curves were calculated from soil water retention data from
the LandIS database (see Table 1). The method used is shown in Methods S10.



As an alternative, pedotransfer functions (PTFs) can be a proxy technique that can be
used to derive these properties from simple to measure soil characteristics such as BD
and soil carbon (C) (Matula *et al.*, 2007; Mayr & Jarvis, 1999). Two types of PTFs were
considered: the first represents a standard type PTF that is derived using multiple linear
regressions (MLRs). The second is an extension of the MLR approach in which
categorical data such as 'Soil Series' and 'Land use' can be considered. Multiple
Additive Regression Splines (MARS) is a nonparametric regression technique that
combines both regression splines and model selection methods (Friedman, 1991). The
general method used for the PTFs is described in Methods S9.
The results of the PTF were compared for fit (Table 3) and show a high level of
agreement. MARS regression approaches tended to perform better than the standard
regression approaches. The predicted values of the SWRC indicators were compared
against the observed values calculated from the Land IS database. Again, there was
good agreement amongst the PTFs (Figure 4) and as such, these approaches are feasible
as a proxy for SWRC. It has been recommended that for a plot of 20 by 20 m, 25
aggregated samples would be required for the measurement of BD and organic C that
are required for the input data for the PTFs (Rickson *et al.*, 2012).
### 3.3    Soil sealing
Soil sealing refers to the impermeabilisation of soils resulting from natural factors
(Pulido Moncada *et al.*, 2014) and human activities (for example road construction)
(Xiao *et al.*, 2013). In the context of this work, soil sealing refers to the covering of soil
surfaces by expanding urban infrastructure.



Soil sealing has been identified as one of the greatest threats to soil functions in the UK
(Rawlins *et al.*, 2013) and worldwide (García *et al.*, 2014; Jie *et al.*, 2002). The growth
of these impervious areas is regarded as an indicator of land degradation (Munafò *et al.*,
2013) as it results in interruptions to gaseous, water and energy exchanges in soils (for
example water regulation), decreased biomass production and increased concentrations
of soil pollutants (Scalenghe & Marsan, 2009). Soil sealing also has a climatic impact
by altering surface albedo and air temperature, and can impact on soil biogeochemical
cycles (Gregory *et al.*, 2015b; Zhao *et al.*, 2012). In order to observe and assess changes
in these soil functions, the change in the proportion of sealed surfaces must also be
monitored.
There are a number of methods to evaluate soil sealing that have been used in the past,
ranging from statistical analysis of national cadastral maps to aerial photo interpretation
(Rickson *et al.*, 2012). Currently, remote sensing techniques are favoured as they have a
large spatial and temporal coverage, have improved certainty of measurements and also
provide base-line data on the proportion of sealed soils within urban areas. The extent of
the built environment can be estimated using a number of remote sensing techniques
including high resolution satellite imagery ( <1 m ground resolution) and aerial
photography. Both these methods allow for the inclusion of narrow corridors such as
roads and rail tracks, as well as providing accurate estimates of unsealed soil areas
surrounding urban areas (i.e. green spaces). By integrating remote sensing with other,
existing databases such as soil maps, even finer spatial resolutions can be achieved
(Rickson *et al.*, 2012).





In terms of measurement, the key indicators for soil sealing are: 1. the absolute area of
sealed soil (ha) and 2. the change/growth rate of area of sealed soil (ha $yr^{-1}$, ha $d^{-1}$, %
change to baseline). With the first indicator, the levels of soil sealing can depend on a
number of factors including policy decisions, individual's choice and the degree of
coverage (Meinel & Hernig, 2005) ranging from 100% sealed (roofs, concrete, asphalt);
70% sealed (paving slabs with seep-able joints); to 50% sealed or less (green roofs,
gravel, crushed stone, porous pavements). The measure of change/growth rate
associated with the second indicator must also incorporate any de-sealing (or negative
sealing) that would reduce the extent of sealed soil (for example installation of green
roofs, porous block paving, porous tarmac and geotextiles used in car parks). This
would require very high resolution ( < 1 m) monitoring data as the areas can be small
and fragmented.
There are a number of earth observation data that can be used (Table 4) for identifying,
classifying and monitoring soil sealing. They all have advantages, disadvantages and
considerations for the user in terms of sampling/ data analysis effort required. One of
the most important considerations is to do with spatial and temporal scale. The use of
remotely sensed information allows population estimates to be made in the imaged area
at the pixel resolution. As such there is usually a trade-off between the resolution and
area that is covered. In terms of the spatial scale for urban areas, very high resolution
data (<1 m) is recommended to monitor smaller sealed and fragmented areas such as
domestic driveways. This scale is also recommended for determining de-sealed surfaces
which tend to be small scale.



Regarding appropriate temporal scales of measurement and monitoring, soil sealing in
urban areas can occur on the timescale of months to years depending on what is being
built. Furthermore, the capture of remote sensing data, in particular very high resolution
imagery usually occurs only every 3-5 years (Rickson *et al.*, 2012) and therefore a
monitoring schedule would have to fit around this. If medium to high spatial resolution
imagery is to be used, sampling could take place annually (data is collected more
frequently and has a larger spatial coverage) (Rickson *et al.*, 2012).
**3.4    General Approach**
The multi-stage approach used in this study proved to be flexible and whilst there was
paucity in the data, it can be altered according to the needs of the end user/monitoring
body/policy maker and what they want to get out of a soil monitoring program. These
diverse needs can be reflected for example in the priorities set in the logical sieve
process, cost considerations, sample numbers and/or what constitutes meaningful
change for that end use. In order to test for meaningful change in the selected indicators,
spatial and temporal data is required to reflect the variability of each property (signal:
noise ratio). However, in the examples given, recommendations for a sampling effort
were given based solely on the scientific literature. In this case, the evidence base was
poor in terms of data that is meaningful (i.e. degree of change in the SQI that will affect
soil processes and functions) and detectable (sample size required to detect the
meaningful signal from the variability in the signal) (Rickson *et al.*, 2012). In order to
overcome this, further work is required to build up the evidence base in terms of spatial
and temporal data on the key SQIs. Where other sampling issues were identified,
suitable proxies or modelling functions were tested and proved to be effective in terms



of how well they correlated to the standard measurements for the indicator and any
time/labour issues associated with its measurement.
**4    Conclusion**
This study has demonstrated a multi-stage process that prioritises and analyses the
suitability of physical SQIs for monitoring soil quality and function. In the first stage a
logical sieve and scenario approach were used to prioritise candidate physical SQIs
from the literature. These were then assessed for uncertainty in their measurement,
spatial variability, expected rate of change and impacts on soil processes and functions.
Of the seven prioritised physical SQIs, three were selected as case studies representing
the varying degrees of analysis and modelling that could be applied depending on the
evidence base.
By emphasising the current key soil functions related to current soil and environmental
policy (i.e. provisioning and regulating functions), the prioritised SQIs can be related to
soil processes, soil functions and consequent delivery of ecosystem goods and services.
These are likely to shape any future soil and environmental policy, as well as efforts to
develop soil monitoring programs that aim to evaluate soil physical quality.
**Acknowledgements**
This research was funded by the UK Department of Environment, Food and Rural
Affairs (DEFRA Project Number: SP1611). Part of this work (TM time) was conducted
as part of the Fragments, Functions and Flows in Urban Ecosystem Services (F[3]UES)
Project (Grant Number NE/J015067/1) with support from the Biodiversity and
Ecosystem Service Sustainability (BESS) programme. BESS is a six-year programme



(2011-2017) funded by the UK Natural Environment Research Council (NERC) and the
Biotechnology and Biological Sciences Research Council (BBSRC) as part of the UK's
Living with Environmental Change (LWEC) programme. It was also conducted under
BBSRC grant BB/M011860/ which covered RC's time and contributions to the paper.
The views expressed are those of the authors and not necessarily those of DEFRA,
NERC or BBSRC.


## Supporting Information

**Table S1.** Collated list of potential physical indicators of soil quality considered in the ranking exercise, with associated sub-categories and indicator numbers. Adapted from Rickson et al. (2012).

**Table S2.** Pertinence of physical soil quality indicators to the 'Soil Functions Category' of the logical sieve, with scoring system. Adapted from the Millennium Ecosystem Assessment (2005).

**Table S3.** Pertinence of physical soil quality indicators to the 'Land Use Category' of the logical sieve, with scoring system. Adapted from CEH (2007).

**Table S4.** Pertinence of physical soil quality indicators to the 'Soil Degradation' of the logical sieve, with scoring system. Adapted from European Commission (European Commission, 2006, Table 4).

**Table S5.** Scoring values allocated to each of the challenge criteria used to evaluate physical soil quality indicators. Adapted from Merrington et al. (2006) and Huber et al. (2008).

**Methods S6.** Methodology for weighting factors, scoring and ranking indicators Adapted from Ritz et al. (2009) and Rickson et al. (2012).

**Table S7.** Example of logical sieve assessment (Physical SQI – Rate of erosion IND11)

**Methods S8.** Geostatistical modelling technique. Adapted from Rickson et al. (2012).

**Methods S9.** Methodology for determining pedotransfer functions. Adapted from Rickson et al. (2012).

**Methods S10.** Methodology for determining soil water retention characteristics (index S, AC, PAWC and RFD) from the LandIS data base. Adapted from Rickson et al. (2012).



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



**Table 1: Datasets and analyses for selected physical SQIs**

| Physical SQI | Available Datasets | Data | Analyses |
|---|---|---|---|
| **Packing density** | LandIS (Soil Survey of England and Wales) – | 1,250 measurement of bulk density and clay content averaged over soil profiles | • Power analysis<br>• Spatial Statistics |
| | ADAS (DEFRA project BD5001) (Price *et al.*, 2012) | 300 short range measurements of bulk density | |
| | DEFRA project SP1606 (Graves *et al.*, 2011) | Supra-classifications of soil/land use combinations | |
| **Soil water retention characteristics** | LandIS (Soil Survey of England and Wales) – | 2,480 soil profiles with soil water retention values. Volumetric moisture content measured at pressure heads of 0.5, 1, 4, 20 and 150 m. Total porosity (%) | • Hydrological modelling<br>• Pedo-transfer functions |
| **Soil sealing** | Remote sensing data | Discussion of available methods to measure and monitor soil sealing. | • Considerations of pixel size and appropriate satellite images for determination of sealing of soil and degree of imperviousness |



**Table 2: Soil water retention characteristics indicators, optimum values and impacts on the provisioning soil function. Values for PAWC, M and RFC taken from Reynolds et al. (2009). Values for Dexter S value taken from (Dexter, 2004a). $\theta_{FC}$ = volumetric moisture content at field capacity, occurring at 0.5 or 1 m pressure head; $\theta_{sat}$ = saturated moisture content at 0m pressure head; $\theta_{PWP}$ = moisture content at permanent wilting point, occurring at 150 m pressure head; $\theta_m$ = porosity of the soil matrix occurring at 0.1 m pressure head.**

| Indicator | Optimum Values | Soil Function (i.e. provisioning function: root growth) | |
|---|---|---|---|
| **Plant Available Water Capacity (PAWC)** $(PAWC = \theta_{FC} - \theta_{PWP})$ (vol / vol; cm³.cm⁻³) | $PAWC \geq 0.20$ | Maximal root growth and function (will vary according to crop type and variety) | |
| | $0.15 \geq PAWC \leq 0.20$ | Good | |
| | $0.10 \geq PAWC \leq 0.15$ | Limited | |
| | $PAWC \leq 0.10$ | Poor for root development | |
| **Macroporosity (M)** $(M = \theta_{sat} - \theta_m)$ (cm³ . cm⁻³) | $M \geq 0.05{-}0.10$ | Optimal | |
| | $M \leq 0.04$ | Soils degraded by compaction | |
| **Relative Field Capacity (RFC)** (rain-fed agriculture and mineral soils) $(RFC = \theta_{FC} / \theta_{sat})$ | $0.6 \leq RFC \leq 0.7$ | Optimal balance between available water and air capacity | |
| | $RFC \leq 0.6$ | Insufficient water - droughtiness | Potential reduction in microbial activity, in particular microbial production of nitrate. |
| | $RFC \geq 0.7$ | Insufficient air - waterlogging | |
| **Dexter S value ($S_g$)** | $S_g < 0.020$ | 'Very poor' soil physical quality | |
| | $0.020 \geq S_g \leq 0.035$ | 'Poor' soil physical quality | |
| | $0.035 \geq S_g \leq 0.050$ | 'Good' soil physical quality | |
| | $S_g \geq 0.050$ | 'Very good' soil physical quality | |





**Table 3: Soil water retention characteristics: Fit results from PTFs based on LandIS data (BD, texture [clay, silt and sand] and organic C content). $S_v$ is related to $S_g$ through the soil bulk density $\rho_b$ $S_v = \rho_b S_g$**

| | $S_v$ | | $S_g$ | | Relative Field Capacity | | Drainable Porosity | | Plant Available Water | |
|---|---|---|---|---|---|---|---|---|---|---|
| | RSQ | conc R | RSQ | conc R | RSQ | conc R | RSQ | conc R | RSQ | conc R |
| **multiple regression** | 0.56 | 0.73 | 0.82 | 0.85 | 0.61 | 0.78 | 0.53 | 0.65 | 0.58 | 0.72 |
| **MARS splines** | 0.72 | 0.75 | 0.87 | 0.9 | 0.73 | 0.85 | 0.71 | 0.08 | 0.68 | 0.82 |

RSQ = $R^2$ statistic; conc R = concordance correlation



**Table 4: Remote sensing datasets for the identification of sealed soil in urban areas. Adapted from Rickson et al. (2012)**

| Datasets | Resolution | Spectral bands | Measurements | Classifications | Advantages | Disadvantages |
|---|---|---|---|---|---|---|
| **Medium-High Resolution Earth Observation (EO) satellite data** | | | | | | |
| **NASA's Landsat** | High resolution (30 m) | Multi spectral (7 bands) | Vegetation Indices: Calculated from sensors with R and NIR sensitive to vegetated (un-sealed) surfaces. The Normalised Difference Vegetation Index (NDVI) is the most widely used. | Classification algorithm: Pixel-based digital classifications (PDC) can be used to automatically characterise the landscape in imagery based on probabilistic pixel level digital image processing. Maximum Likelihood algorithm used. | Urban areas can be easily separated due to large spectral differences between vegetation and urban infrastructure. | For medium resolution imagery, sealed areas < 1-2 times the pixel area cannot be resolved. |
| **Disaster Monitoring Constellation (DMC)** | High resolution (2.5 – 32 m) | Multi spectral (3 bands) | | | | |
| **SPOT 5 imagery** | High resolution (10m) | Multi spectral (3 bands) | | | | |
| **Very High Resolution (VHR) Earth Observation data** | | | | | | |
| **Very high resolution satellite imagery** | Very high resolution (<5 m) | Multispectral | A pixel classifier is not suitable in this case as it results in an increased variation in the statistical definition of a 'building' class and decreases classification accuracy. | Alternative image analysis techniques: Object based classifiers using semi-automated classification. Aerial Photo Interpretation (API) is used to manually edit and then classify objects before classification is run on the EO data. | Can be useful when considering a smaller spatial scale than with medium resolution imagery. | PDC less effective at this scale due to the pixel size. Texture effects are visible and the spectral response of a cover class is disaggregated. |
| **Digital aerial photography collections** | Very high resolution (<1 m) | Not Applicable | | | | |
| **Other Remote Sensing Datasets** | | | | | | |
| **LIDAR (Light detection and ranging)** | Very high resolution (<1 m) | Laser (~1550 nm) | Determine the elevation of buildings and vegetation canopies. | Digital classification from differences in surface types from height for Lidar. | Very accurate elevation, possible to have terrain and surface models depending on look angle. | Flown at low level, with slow aircraft, covering small areas. |
| **SAR (Synthetic Aperture Radar)** | Very high resolution (<1 m) | Microwave (cm), different polarizations, (e.g. VV, HH, VH, HV) | Determine the surface types below tree canopies (and can measure through cloud). | Digital classification or surface types from Backscatter coefficient for SAR. | Polarimetric SAR useful for separating urban and natural vegetation, SAR also useful for detecting water bodies (low backscatter coefficients), and can measure through cloud. | Radar speckle (noise) removal, terrain displacement possible in hilly areas. |





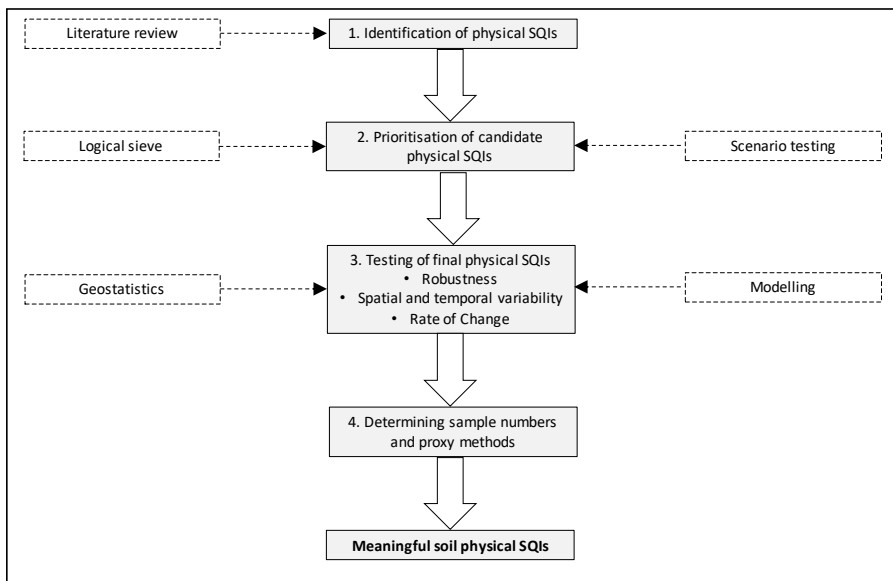

**Figure 1: Multi-stage approach taken in the selection of meaningful physical soil quality indicators (SQIs)**





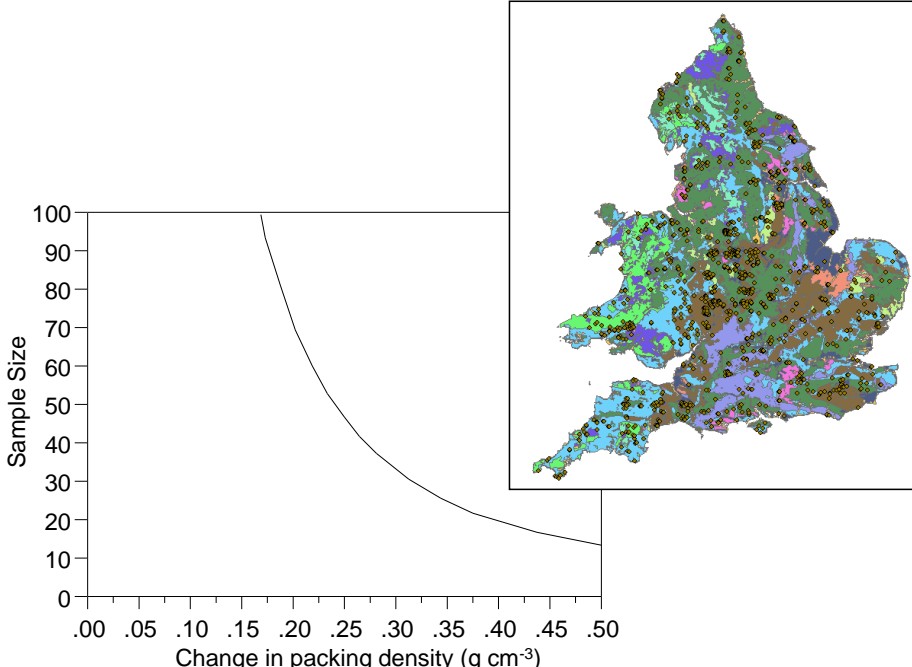

**Figure 2: Power analysis on national soil packing density data based on land use by soil strata. The spatial distribution of the data points superimposed on the land use /soil classification is taken from** (Graves *et al.*, 2011)

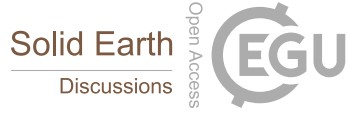

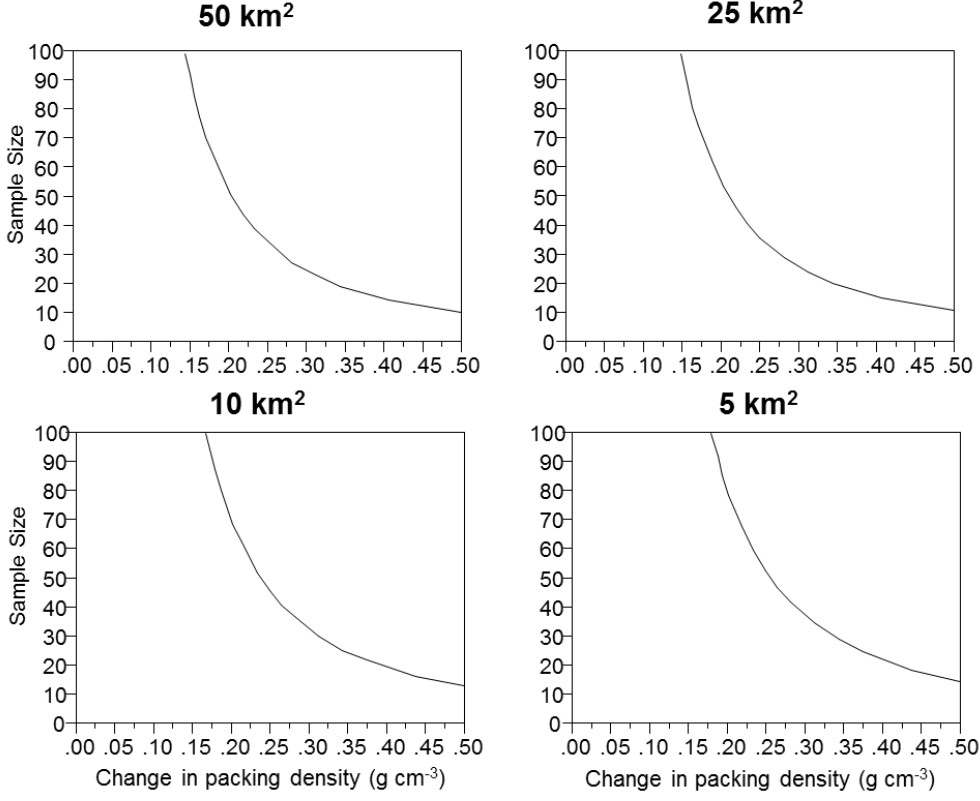

**Figure 3: Power Analysis using a model based approach in which the variability was estimated given a particular block size using the variogram described in Methods S8 .Soil water retention characteristics**







**Figure 4: Biplots representing the predicted SQI versus observed SQI based on the MARS pedotransfer functions. a) Relative Field Capacity, b) Drainable Porosity, c) Plant Available Water**