# Peer review of "Corstanje et al. [Land Degradation and Development] Table S1. Collated list of potential physical indicators of soil quality considered in the ranking exercise, with associated sub-categories and indicator numbers. Adapted from Rickson et al. (2012)."

_Solid Earth, 2016_

## Short Comment (SC1) · 21 Nov 2016

Congratulations for your nice work. I just suggest redesigning Fig. 4 and use one-column format. May you change yellow into white color?

Antonio Jordán Executive Editor of Solid Earth

---

## Referee Comment (RC1) · Anonymous Referee #1 · 4 Dec 2016

Solid Earth Discuss., doi:10.5194/se-2016-153, 2016 Manuscript under review for journal Solid Earth: PHYSICAL SOIL QUALITY INDICATORS FOR MONITORING BRITISH SOILS

The manuscript deals with the selection of physical soil indicators suitable for monitoring activities in England. The subject is topical and interesting also beyond the UK, but it has been not treated properly. Main drawbacks are: i) the section dealing with soil sealing is somehow and attachment to that dealing with soil qualities. It should have been treated in a separate paper, possibly. On the other hand, the first part, related to soil qualities, could be expanded ii) the text is difficult to follow because of a) too many relevant supplementary materials, b) literature difficult to find (a major methodological issue is addressed to Rickson et al.,2012), c) same concepts expressed in different ways iii) the authors probably assume that the manuscript focuses on soil qualities related to the topsoil of agricultural lands, however, soil depth is also mentioned. On the other hand, physical properties of the underlying soil horizons are not mentioned. iv) I am not sure that block sized 5 or 10 km2 approximate management units of field size, this point should be clarified v) Results should be separated from discussion

Other pitfalls are: 1. pag 2: In the introduction, the role of soil as a cultural heritage has not been treated. Actually, although soil is often considered as container of archaeological findings, its other not monetary values, e.g., among others, convener of information about events occurred during historical and geological times, should be much more acknowledged. Nevertheless, there are several publications on this issue, and even databases, which could be mentioned. 2. pag 3: indicators of soil quality are also required to assess the effectiveness of restoration strategies, some recent papers on solid earth well illustrated this issue 3. pag 4: the relationship between application of sensor technology and indicators could be better treated, since it is on cutting edge and treated by several authors 4. pages 8 and 9: calling same things with different names, i.e., Category and Criteria, functions and factors, creates some confusion in the reader 5. lines 219-238: this section should be expanded, providing for details and examples 6. line 252: what do you mean with power analysis? 7. line 313: in fig 3 it is very difficult to detect the stated differences in sample size. May be using one graph only? 8. table 3: specify Sv

---

## Short Comment (SC2) · 18 Dec 2016

This study aimed at a systematic process to select physical SQI and how these indicators be applied in large scale. This study provides useful information for enriching knowledge in corresponding fields. Suggestions as follows: Line 281: what is clay diameter range? May be you can added the range like Line 287. Page 34, In Figure3, the four figures can put into a figure.

---

## Referee Comment (RC2) · Anonymous Referee #2 · 1 Jan 2017

The methods including the three steps to selecting SQI should be clearly stated.

Why were only three SOIs been discussed? how about others?

Was six or seven SQIs were determined at last?

Some critical tables should be listed in the manuscript instead of in supplementary categories.

How to use these selected SQIs to quantify British soil quality should be added to the manuscript.
* * *

---

## Author Comment (AC1) · 21 Feb 2017

Thank you for your kind comments. We have now redesigned Fig. 4 using a one column format. The background has been changed from yellow to white.

---

## Author Comment (AC2) · 21 Feb 2017

The manuscript deals with the selection of physical soil indicators suitable for monitoring activities in England. The subject is topical and interesting also beyond the UK, but it has been not treated properly. Main drawbacks are:

i) The section dealing with soil sealing is somehow an attachment to that dealing with soil qualities. It should have been treated in a separate paper, possibly. On the other hand, the first part, related to soil qualities, could be expanded.
Rather than an add-on or attachment, soil sealing was identified through the robust multi-stage approach using the literature, logical sieve and scenario-based approach. As such, the final physical SQIs were supported by the literature and also by the common needs of stakeholder groups. Whilst the approach to measuring soil sealing may seem very different to the other physical SQIs, it is nonetheless significant and we felt that it was important to include it as an example in contrast to the other physical SQIs. The three selected SQIs that we included in the manuscript were selected to illustrate the different approaches to the types of analyses that can be conducted depending on the type of soils data available (i.e. quantitative methods such as power analysis or pedo-transer functions; semi-qualitative such as remote sensing).

ii) The text is difficult to follow because of:
   a. Too many relevant supplementary materials,
   We have attempted to include the bare minimum in terms of supporting figures (four in total) and tables (four in total) in the main text of the manuscript. The supplementary materials are there largely for reference purposes for anyone wishing to replicate the approach that we took. They don't necessarily need to be referred to in order to follow the manuscript.

   b. Literature difficult to find (a major methodological issue is addressed to Rickson et al.,2012),
   Where possible we have included scientific papers that can be accessed through subscription. Some of the datasets used such as the CEH Land Cover Map require licences to access them and where this is the case, we have included the relevant information in the manuscript to ensure that it doesn't need to be referred to by the readership. The reports referred to in the text (such as DEFRA and the Rickson et al., 2012 paper) are all freely available online.

   c. Same concepts expressed in different ways
   We attempted to clarify a few terms that may be confusing. For example, in the logical sieve, the terms 'criteria' 'categories', 'functions' and 'factors' have been used interchangeably in some instances. This has been addressed and the terms have been more explicitly defined and explained with examples for clarification.

iii) The authors probably assume that the manuscript focuses on soil qualities related to the topsoil of agricultural lands, however, soil depth is also mentioned. On the other hand, physical properties of the underlying soil horizons are not mentioned.
The physical SQIs were evaluated according to four criteria, one being land use. In this category, the candidate SQI should apply to all land uses found nationally including those resulting from different land management practices. As such, we do not solely refer to the topsoil of agricultural lands. The depth of soil was only mentioned in the section on packing density (page 13) in its use to measure bulk density of soils using the Kopecky ring method. We acknowledge that the depth of soil was mentioned as one of the final physical SQIs for further analysis, however, we selected three examples from the final shortlist (packing density; soil water retention characteristics; soil sealing) and this is why the depth of soils and underlying soil horizons is not addressed in the text. However, these are covered in the Rickson et al. 2012 Final report and WP2 report available online.

iv) I am not sure that block sized 5 or 10 km$^2$ approximate management units of field size, this point should be clarified.
We have clarified this in the text where 5 and 10km$^2$ refers to field scale, 25km$^2$ refers to farm scale and 50km$^2$ refers to landscape level.

v) **Results should be separated from discussion.**
We have referred to a number of research papers in *Solid Earth* for formatting and style. There are quite a number of papers that have an integrated result and discussion section. We felt this approach more appropriate as it meant that we were integrating our findings, thereby preventing repetition in the text and keeping the word count down.

vi) Other pitfalls are:
1. pag 2: In the introduction, the role of soil as a cultural heritage has not been treated. Actually, although soil is often considered as container of archaeological findings, its other not monetary values, e.g., among others, convener of information about events occurred during historical and geological times, should be much more acknowledged. Nevertheless, there are several publications on this issue, and even databases, which could be mentioned.
The introduction has now been amended to include soil cultural heritage:

In recent years soil quality and its measurement have increasingly been based on soil functions (Loveland & Thompson, 2002; Ritz *et al.*, 2009; Rosa, 2005). These functions determine the ability of a soil to deliver and support ecosystem goods and services, which have been linked to human health and wellbeing. Soils are typically recognised for their role in provisioning goods such as building material, fresh water, fuel, fibre and food (Robinson *et al.*, 2013). They also interact with other environmental components (air and water) and provide a platform for infrastructure. The ecosystem services that rely on these functions include regulation of climate and hydrology, contaminant transformation, biocontrol of plant pathogens and parasites (Sylvain & Wall 2011) and water filtration/runoff reduction/purification (Breure *et al.*, 2012). Supporting services provided by soils include soil formation, soil fertility, biogeochemical cycling (C storage and nutrient cycling), decomposition of organic materials and plant available water. A number of cultural services are also supported such as recreational surfaces (Robinson *et al.*, 2013) and heritage services such as preserving historic artefacts, burial grounds and non-monetary values such as containing information of events that have occurred in historical and geological timescales (Costantini & L'Abate, 2009). There are a number of databases and protection in place for soils with cultural heritage (otherwise known as pedosites, geosites and geoparks) (Geosite, 2017; UNESCO, 2016).

**2. pag 3: indicators of soil quality are also required to assess the effectiveness of restoration strategies, some recent papers on solid earth well illustrated this issue**
**The text on page 3 has now been amended to include the role of soil quality indicators in assessing the effectiveness of restoration strategies:**
In order to measure soil quality and function, soil quality indicators are commonly used. Indicators of soil quality are required for environmental monitoring/reporting and provide the basis for many soil protection policies and monitoring programs (Pulleman *et al.*, 2012). They help assess human and natural impacts on soils and to identify the effectiveness (or otherwise) of sustainable land management practices (Doran & Parkin, 1994; Karlen & Stott, 1994; Schipper & Sparling, 2000). They have also been used to assess the effectiveness of restoration strategies (Costantini *et al.*, 2016).

**3. pag 4: the relationship between application of sensor technology and indicators could be better treated, since it is on cutting edge and treated by several authors**
**We briefly mentioned the use of sensor technology in the practical aspects of monitoring soil (page 4). We then go onto discuss the benefits and relationship between sensor technology and indicators (sample size) in the section on soil sealing (backed up with recent studies) as well as in the packing density section (see excerpt below):**
A number of proxies exist that overcome some of the issues regarding sampling effort using the traditional Kopecky ring method. These allow a higher resolution of measurements (1500-2500 samples) per hectare over larger areas and include on-line (mobile) and non-mobile systems (Rickson *et al.*, 2012). The methods used require multiple sensors and advanced techniques for data analysis (Mouazen & Ramon, 2006) such as a combination of Visual and Near Infrared (vis-NIR) measurements, combined with Theta probe determinations for soil moisture or with soil resistance (penetrometer measurements) and vis-NIR measurements to determine BD (and thus PD, when combined with clay content).

**4. pages 8 and 9: calling same things with different names, i.e., Category and Criteria, functions and factors, creates some confusion in the reader**
**We attempted to clarify a few terms that may be confusing. For example, in the logical sieve, the terms 'criteria' 'categories', 'functions' and 'factors' have been used interchangeably in some instances. This has been addressed and the terms have been more explicitly defined and explained with examples for clarification. These are also explained more in the supplementary materials.**

**5. lines 219-238: this section should be expanded, providing for details and examples**
**This section is largely addressed in the Rickson et al. (2012) study which we refer to in the text for reference:**
These were then further filtered by the project team in the Rickson et al. (2012) study to ensure the results of the logical sieve exercise were sensible and no indicators were disqualified unduly, data were available to test the robustness of the selected SQIs, duplication/surrogacy and scale issues (i.e. upscaling) were considered.

**6. line 252: what do you mean with power analysis?**
Statistical power is the probability that a specific difference will be detected at a specified level of confidence. It allows the determination of a sample size required to detect an effect of a given size with a degree of confidence. **This has been added to the text.**

**7. line 313: in fig 3 it is very difficult to detect the stated differences in sample size. May be using one graph only?**
**The four graphs in Figure 3, page 33 have been presented alongside each other to ease the comparisons of relationships between them according to block size.**

**8. table 3: specify Sv**

$S_v$ is related to $S_g$ through the soil bulk density $\rho_b$ $S_v = \rho_b\ S_g$.

---

## Author Comment (AC3) · 21 Feb 2017

Interactive Comment 2:

L. Wang (wlhsoilwater@nwafu.edu.cn)

This study aimed at a systematic process to select physical SQI and how these indicators be applied in large scale. This study provides useful information for enriching knowledge in corresponding fields. Suggestions as follows: Line 281: what is clay diameter range? May be you can add the range like Line 287. Page 34, In Figure3, the four figures can put into a figure.

Thank you for your useful comments. The range given in line 287 refers to number of samples taken, rather than diameter range of the samples measured. As such, we are

uncertain as to what range the reviewer is addressing. Clay content was obtained as part of the National Soils Inventory Database (NSI), details of the analytical methods can be found in http://www.landis.org.uk/data/nsi.cfm

The three figures in Figure 4, page 34 have been redesigned into a single column format. The four graphs in Figure 3, page 33 have been presented alongside each other to ease the comparisons of relationships between them according to block size.

---

## Author Comment (AC4) · 21 Feb 2017

i. **The methods including the three steps to selecting SQI should be clearly stated.**
   **We have stated the three steps in the final paragraph of the introduction and at the start of the methodology. Figure 1 was also designed to clearly illustrate the approach taken:**

This study uses a multi-stage approach in the selection and prioritisation of physical SQIs that meet the required criteria and conditions outlined above. It consists of a systematic review and selection procedure, followed by assessment of the selected SQIs and how they could be best applied at a National scale monitoring programme. The final priority list should indicate the soil's capacity to deliver ecosystem goods/services and are therefore indicative of soil quality.

The process of physical SQI selection takes a multi-stage approach as outlined in Figure 1. In the first stage, potential physical SQIs were identified from the available literature, including those defined by Loveland and Thompson (2002) and Merrington et al. (2006). Other physical SQIs (and the methods used to measure them) that had not been considered previously were also included to produce an up-to-date list. In the second stage, the candidate physical SQIs were prioritised using a logical sieve (Ritz *et al.*, 2009) and a scenario-based approach. In this approach, the logical sieve was interrogated by running three scenarios based on typical priorities of different stakeholders by applying weightings to the scores. As such, the approach was be used to prioritise a specific soil function or degradation process of interest (Rickson *et al.*, 2012).

Finally, the priority physical SQIs were tested for robustness (statistical reliability and accuracy as well as practicability), spatial and temporal variability and expected rate of change using statistical analysis and modelling. This involves determining appropriate sample numbers for defining meaningful change as well as proxy methods that can be used to make the physical SQI measurements operational and feasible. For example, where a standard measurement physical SQI measurement may be time or resource intensive to measure and monitor in a large scale monitoring programme, an easier/cheaper to measure proxy may exist that could make that physical SQI feasible for inclusion into such a programme.

[Figure]

**Figure 1: Multi-stage approach taken in the selection of meaningful physical soil quality indicators (SQIs)**

ii. **Why were only three SOIs been discussed? how about others? Was six or seven SQIs were determined at last?**
**The process actually identified 7 SQIs that met all of the criteria discussed. These can be referred to in the Rickson et al. (2012) report which is freely available online. However, we only selected three SQIs to include in the manuscript to illustrate the different approaches to the types of analyses that can be conducted depending on the type of soils data available (i.e. quantitative methods such as power analysis or pedo-transer functions; semi-qualitative such as remote sensing). The selection was also important in order to keep to the word count.**

iii. **Some critical tables should be listed in the manuscript instead of in supplementary categories.**
**We have included four supporting figures and four supporting tables in the main text of the manuscript. It was felt that including too many more would detract from the main argument of the text and therefore we have included critical information in the supplementary materials for reference purposes. However, they don't necessarily need to be referred to in order to follow the manuscript.**

iv. **How to use these selected SQIs to quantify British soil quality should be added to the manuscript.**
**The text has now been amended in the conclusion section:**
By emphasising the current key soil functions related to current soil and environmental policy in the UK (i.e. provisioning and regulating functions), the prioritised SQIs can be related to soil processes, soil functions and consequent delivery of ecosystem goods and services. These are likely to shape any future soil and environmental policy in the UK, as well as efforts to develop soil monitoring programs that aim to evaluate soil physical quality.